# Synthesis of Microporosity Dominant Wood-Based Activated Carbon Fiber for Removal of Copper Ions

**DOI:** 10.3390/polym14061088

**Published:** 2022-03-09

**Authors:** Zhi Jin, Zhen Zeng, Shenghui Hu, Lina Tang, Yuejin Fu, Guangjie Zhao

**Affiliations:** 1Research Institute of Wood Industry, Chinese Academy of Forestry, Beijing 100091, China; 13661011605@163.com (Z.J.); z13522948894@163.com (Z.Z.); yihui648@foxmail.com (S.H.); 18211090798@163.com (L.T.); 2College of Material Science and Technology, Beijing Forestry University, Beijing 100083, China

**Keywords:** Chinese fir, activated carbon fiber, steam activation, porosity, Cu^2+^ adsorption

## Abstract

Steam activation treatments were introduced in the preparation of activated carbon fiber from liquefied wood (LWACF), to enlarge its specific surface area and develop the pore size distribution. With increasing activation time, the average fiber diameter of LWACF decreased from 27.2 µm to 13.2 µm, while the specific surface area increased from 1025 to 2478 m^2^/g. Steam activation predominantly enhanced the development of microporosity, without significant pore widening. Prolonging the steam activation time exponentially increased the removal efficiency of Cu^2+^ at a constant adsorbent dose, as a result of an increase in the number of micropores and acidic-oxygenated groups. Moreover, for LWACF activated for 220 min at 800 °C, the removal efficiency of Cu^2+^ increased from 55.2% to 99.4%, when the porous carbon fiber dose went from 0.1 to 0.5 g/L. The synthesized LWACF was proven to be a highly efficient adsorbent for the treatment of Cu^2+^ ion-contaminated wastewater.

## 1. Introduction

With the rapid development of industrial activities, numerous effluents containing heavy metals are released into surface and underground water, resulting in increased environmental risks. Copper is a heavily used metal in industries such as plating, mining and smelting, brass manufacture, electroplating industries, and petroleum refining, and excessively used in Cu-based agrichemicals mining [1,2,3]. These industries produce much wastewater and sludge containing Cu^2+^ ions at various concentrations, which have negative effects on the water environment [4,5]. It has been reported that heavy metals such as lead, mercury, and copper are toxic and non-biodegradable, adversely impacting human health and causing central nervous system problems. Therefore, developing effective technologies to treat Cu^2+^ polluted wastewaters is urgent, before their discharge into natural environment.

The most commonly used techniques for Cu^2+^ ion removal from aqueous solutions include oxidation, reduction, precipitation, membrane filtration, biological process, ion exchange, and adsorption [6,7,8,9]. Among the various treatment technologies, carbon-based porous materials are commonly used, due to their high surface area, abundant surface groups, harmlessness to the environment, and ease of operation. Activated carbon fiber (ACF) is believed to be the most promising porous carbon material in the field of environmental protection [10,11,12]. Commercially, ACF is synthesized using petroleum-based precursors, such as pitch, polyacrylonitrile, viscose rayon, and phenol resin [13,14,15]. To ease petrochemical resource shortages, ACF from liquefied wood (LWACF) has been synthesized successfully and used as supercapacitor electrodes, potential adsorbents for emerging organic contaminants, and catalysis [16,17,18,19,20]. However, its utilization in heavy metal adsorption has not yet been reported, partly due to the difficulty in pore size distribution (PSD) adjustment. Considering the diameter of hydrated Cu^2+^ of 0.144 nm, a reasonable design of LWACF is necessary. At present, the preparation and utilization of mesoporous LWACF with H_3_PO_4_ and ZnCl_2_ activation is widely reported as matrix phase [21], while the investigation on microporous LWACF for heavy metal adsorption has been limited.

Steam activation has emerged as an important method for carbonaceous activation and can improve the PSD of carbon materials. In this respect, some authors reported that steam activation produced carbons with a narrower and more extensive micropore structure than carbon dioxide activation [22,23]. This effect was attributed to the higher diffusion rates and greater accessibility to small pores of the water molecule, due to the smaller size. Meanwhile, steam activation favored the introduction of acidic oxygen containing functional groups during the post-treatment process and the formation of abundant oxygen defects, both of which are regarded as active sites for adsorption [24].

The present work mainly focuses on the preparation of activated carbon fiber made from liquefied wood by steam activation, and the analysis of the impacts of different preparation conditions on the pore structures. The parameters for preparation of LWACF were examined by employing single factor analysis experiments, and the adsorption capacities of this adsorbent were investigated in static adsorption experiments. The findings will promote the wide utilization of biomass-based activated carbon fiber in the field of environmental protection.

## 2. Materials and Methods

### 2.1. Sample Preparation

Oven-dried (75 °C for 12 h) Chinese fir (*Cunninghamia lanceolata* (Lamb.) Hook., (Fu Jian, China) was pulverized and sieved to a particle size of 20–80 meshes to prepare the precursor fibers, through a series of processes including liquefaction, melt spinning, and curing, in order to gain liquefied wood precursor fiber (LWPF) [20]. The LWPF were activated at 800 °C for 60,140, and 220 min by introducing water stream mixed with a nitrogen stream. Finally, they were cooled to room temperature in a nitrogen stream. The synthesizing route of the series of LWACF is presented in Figure 1. The LWACF samples were labeled as ‘LWACF-activation time (min)’. For example, LWACF-60 corresponding to LWACF was prepared by an activation stage at 800 °C for 60 min.

### 2.2. Physicochemical Characterization

To investigate the surface chemistry, XPS measurements were performed on an ESCALAB 250Xi spectrometer (Thermo Fisher Scientific, Waltham, MA, USA) using a monochromatic AlKα X-ray (1486.6 eV) source. The morphology of the synthesized LWACF was characterized by scanning electron microscopy (SEM) (Hitachi S-3400N, Tokyo, Japan). Pore structure of LWACF was analyzed by using an Autosorb iQ (Boynton Beach, Florida, USA) volumetric adsorption analyzer at −196 °C. The Brunauer-Emmett-Teller (BET) model was used to calculate the specific surface area (S_BET_) and average pore size (D_a_). Micropore specific surface area (S_mi_) was obtained by the t-plot method, while the micropore volume (V_mi_), mesopore volume (V_me_), and PSD were determined by density functional theory (DFT).

### 2.3. Cu (II) Adsorption Performance

The Cu^2+^ solutions (pH = 5.5) were prepared by dissolving 0.078125 g CuSO_4_·5H_2_O in 1000 mL distilled water to prepared the required initial concentration 20 mg L^−1^. For kinetic studies, 100 mL of Cu^2+^ solution of known initial concentration and initial pH was taken in a 250 mL screw-cap conical flask with a fixed adsorbent dosage and agitated in a thermostated rotary shaker at a speed of 300 rpm at 25 ± 5 °C for 24 h, to achieve a state of equilibrium, which was far longer than the time to achieve an equilibrated system for metals in previous studies. At various adsorbent dosage intervals, conical flasks were withdrawn and the mixtures were subsequently filtered through 0.22 μm pore size nylon membrane filters (GE cellulose nylon membrane). Heavy metal ions concentrations in the filtrates were determined using a Perkin-Elmer Analyst 800 atomic absorption spectrophotometer (AAS, Perkin-Elmer, Norwalk, CT, USA). The copper hollow cathode lamp was run at current 4.0 mA, and the wavelength was 324.8 mnm. The flame composition was acetylene (flow rate: 2.0 L/min) and air (flow rate: 13.5 L/min).

The removal efficiency (*Q*, %) of the biosorbent on the metal in the solution was determined by the following equation:$$Q=\frac{\left(C_{0}-C_{e}\right)}{C_{0}}\times 100\%$$
where *C*_0_ and *C_e_* are the initial and equilibrium concentration of heavy metal ion in solution (mg/L), respectively.

## 3. Results

### 3.1. Morphology

The SEM images in Figure 2 show the surface morphology of the series of LWACF. It is clear that the surface characteristics of LWACF display clear morphological variations after different activation process. As the activation time increased, the LWACF began to show more pores on its surface and a wider distribution (Figure 2A,C,E). Furthermore, the average fiber diameter of LWACF decreased from 27.2 µm to 13.2 µm, indicating that in parallel with the promotion in pore development, steam activation accelerated the fiber surface erosion. Comparatively, many obvious cracks occurred on the surface and inner core of LWACF-220 (Figure 2F), while for LWACF-60 and LWACF-140 the fiber surface were relatively smooth (Figure 2B,D), suggesting the deeper penetration of steam with prolonged activation time.

### 3.2. Pore Structure

It was observed that LWACF had a typical I adsorption curve characteristic (IUPAC classification) showing a highly microporous nature (Figure 3A,B), and the corresponding porosity parameters derived from the nitrogen isotherms are illustrated in Table 1. The obtained LWACFs displayed a higher BET specific surface area (S_BET_) than the reported H_3_PO_4_ and steam activated carbon fiber [21,22], which increased from 1205 to 2478 cm^3^/g with increasing activation time. It was clearly established that the specific surface and the pore volume in LWACFs increased with the duration of the activation. Indeed, a longer residence time allows the steam to better react with the entire surface, even penetrating into the fiber and improving the creation and development of new pores, resulting in an increase in specific surface and pore volume. Moreover, it was noted that as the activation time prolonged from 60 to 220 min, both S_mi_ and V_mi_ continuously increased and attained a maximum by 220 min of activation. In this process, D_a_ also increased from 1.728 nm to 1.922 nm. Obviously, the activation treatment promotes the development both of microporosity and partial mesoporosity simultaneously, with the former dominant (Figure 3B).

### 3.3. Surface Chemistry

Information about the composition of LWACFs and valence states of their component elements was further analyzed by surface-sensitive high-resolution XPS. The survey spectrum (Figure 4A–C) shows peaks characteristic of C1. It is clear that the C1s binding energies were 284.66–284.74 eV, 284.87–285.26 eV, 286.00–286.40 eV, 288.00–288.91 eV, and 291.17-291.54eV (Figure 3B), which correspond to graphite, the carbon present in phenolic, alcohol, ether, or C=N groups, carbonyl, carboxyl, and carbonate groups, respectively [25,26]. In addition, the curve-fitting results for all samples indicated an increase in the number of oxygen containing groups and a compensatory decrease in the graphite carbon with the increase in the steam activation time (Table 2). Moreover, it was found that increasing the activation time caused an increase of C1s percentage, from 86.30% to 92.29%, while the percentage of O1s decreased from 11.9% to 7.02% (Table 3). Moreover, the O1s/C1s composition ratio decreased with steam activation treatment. This is mainly attributed to the higher content of intrinsic oxygen elements released from the carbon fiber without activation treatment than the oxygen elements introduced by the steam activation (Table 3).

### 3.4. Cu^2+^ Adsorption

The performance of LWACF with regard to Cu^2+^ adsorption was evaluated as a function of sorbent dosage. Adsorbent dosage is an important parameter, because it determines the capacity of an adsorbent for a given initial concentration of the adsorbate. As shown in Figure 5, the removal efficiency of Cu^2+^ was improved with increased activation time at a constant adsorbent dose, which was mainly due to the increase in the S_BET_, V_mi_, and D_a_. In addition to the improved pore structure, the introduction of more oxygen-containing groups also contributed to the enhancement of the removal efficiency of Cu^2+^. It is commonly believed that there are five primary mechanisms of heavy metal removal from aqueous solutions using porous carbon materials; i.e., complexation, cation exchange, precipitation, electrostatic interactions, and chemical reduction [27,28,29]. In the present work, the controlled steam activation of the carbon surface introduced acidic functional groups such as carboxyl and carbonyl, as well as making the carbon surfaces hydrophilic, which promoted the strong attractive interactions between the Cu^2+^ and the active surface of the LWACF.

Thus, there was a synergistic mechanism of chemical complexation and physical adsorption for the higher adsorption capabilities and flash removal rate of the synthesized LWACF. Moreover, it was revealed that the removal efficiency of Cu^2+^ increased from 43.6% to 66.1% for LWACF-140 and from 55.2% to 99.4% for LWACF-220, when the porous activated carbon fiber dose went from 0.1 to 0.5 g. By comparison, the removal efficiency of Cu^2+^ did not show an obvious change for LWACF-60, and even slowed down slightly, when the adsorbent dosage content was increased, which was probably due to the poorly developed porous structure.

## 4. Conclusions

The aim of this study was to develop a series of LWACF samples for the removal of Cu^2+^ ions from an aqueous solution. The effects of steam activation pretreatment were characterized, to elucidate the resulting samples’ adsorptive properties toward heavy metal removal. The removal efficiency of LWACF for Cu^2+^ ion was enhanced with prolonged activation time, possibly due to a simultaneous increase in the number of acidic-oxygenated functional groups and number of micropores. The maximum removal efficiency reached 99.4%. Although 100% metal ion adsorption was not attained at these carbon dosages, the results give a good picture of the utility of LWACF as a metal ion adsorbent.

## Figures and Tables

**Figure 1 polymers-14-01088-f001:**
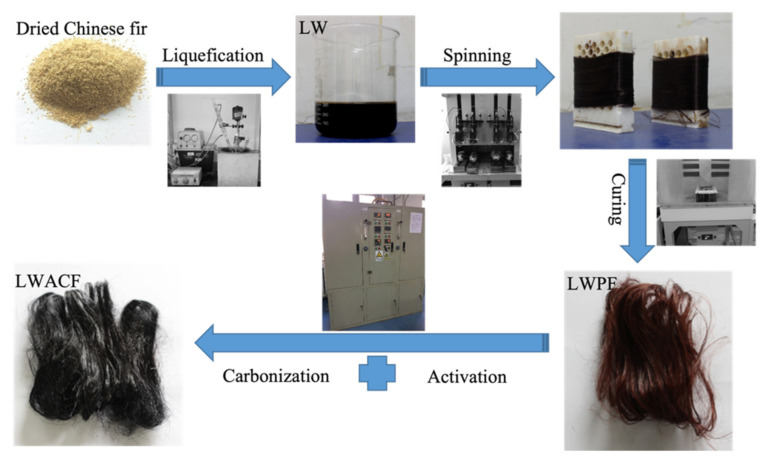
Synthesizing route of the series of LWACF.

**Figure 2 polymers-14-01088-f002:**
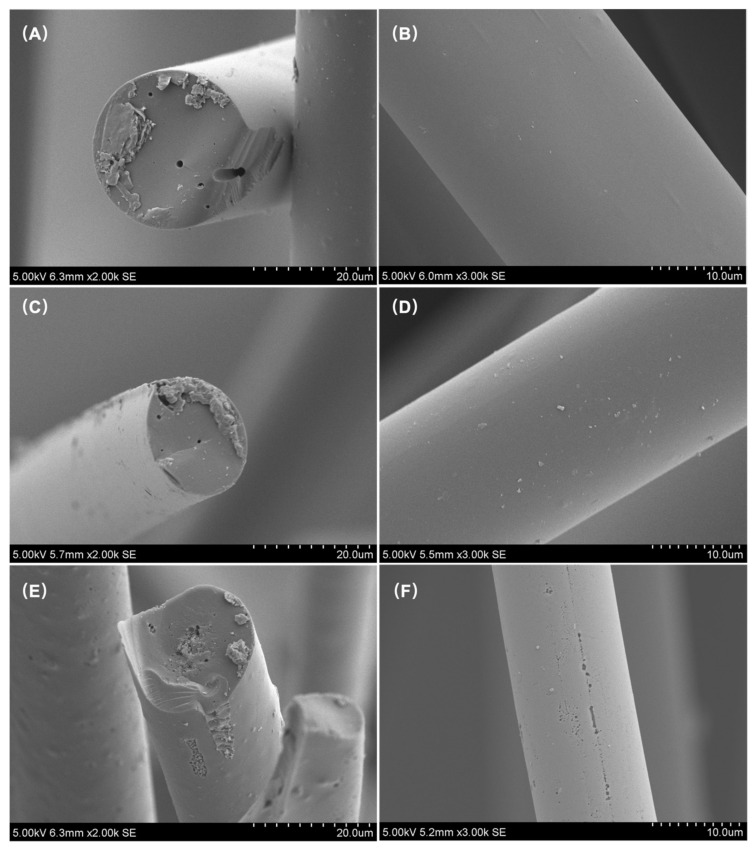
SEM images of LWACFs activated for 60 min (**A**,**B**), 140 min (**C**,**D**), and 220 min (**E**,**F**) at 800 °C.

**Figure 3 polymers-14-01088-f003:**
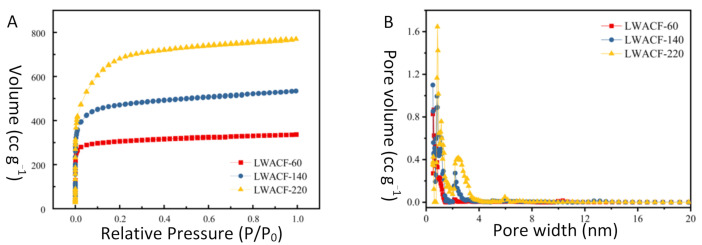
N_2_ adsorption isotherm (**A**) and pore size distribution (**B**) of LWACF.

**Figure 4 polymers-14-01088-f004:**
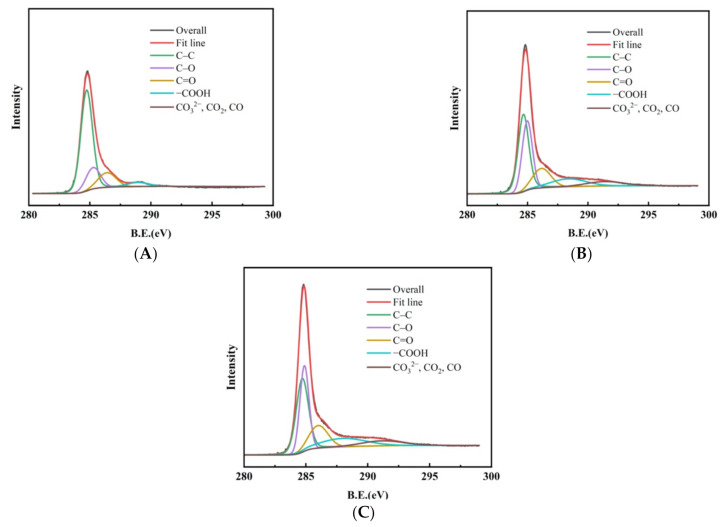
Peak fitting of C1s region of LWACFs activated for 60 min (**A**), 140 min (**B**), and 220 min (**C**) at 800 °C.

**Figure 5 polymers-14-01088-f005:**
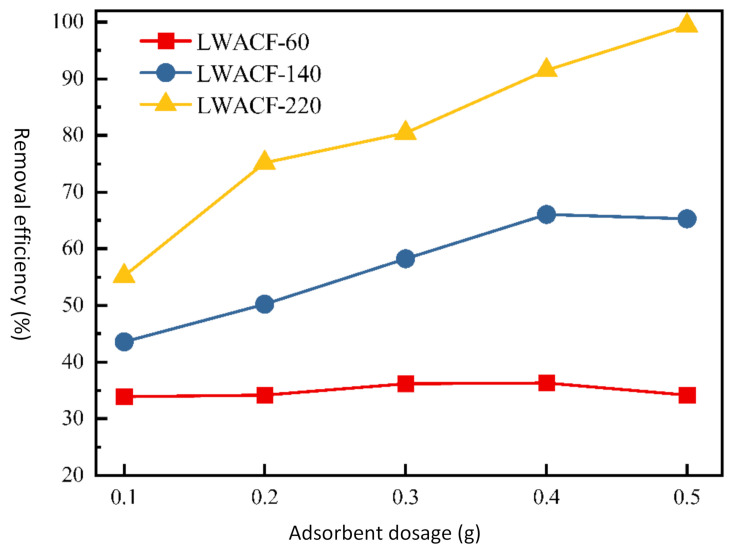
The removal efficiency of Cu^2+^ at a constant adsorbent dose for LWACFs with different activation times at 800 °C.

**Table 1 polymers-14-01088-t001:** Pore structure parameters of LWACFs used for Cu^2+^ adsorption property test.

Samples	S_BET_ (m^2^ g^−1^)	S_mi_ (m^2^ g^−1^)	V_t_ (cm^3^ g^−1^)	V_mi_ (cm^3^ g^−1^)	V_me_ (cm^3^ g^−1^)	S_BJH_ (m^2^ g^−1^)	D_a_ (nm)
LWACF-60	1205	1107	0.521	0.427	0.056	42	1.728
LWACF-140	1801	1604	0.826	0.573	0.189	79	1.835
LWACF-220	2478	2134	1.191	0.652	0.447	94	1.922

**Table 2 polymers-14-01088-t002:** Relative surface concentrations of carbon species obtained by fitting the C1s XPS spectra for LWACFs with different activation times at 800 °C.

Samples	C–C (%)	C–O (%)	C=O (%)	–COOH (%)	CO_3_^2−^, CO_2_, CO (%)
LWACF-60	65.23	16.38	13.72	4.67	0
LWACF-140	37.53	29.37	14.71	11.61	6.78
LWACF-220	34.67	28.30	15.54	13.96	7.53

**Table 3 polymers-14-01088-t003:** Elemental compositions of the surface of LWACFs with different activation time at 800 °C.

Samples	C1s (at%)	O1s (at%)	N1s (at%)	P1s (at%)	S1s (at%)	(O)/(C)
LWACF-60	86.30	11.9	0.77	0.34	0.69	0.138
LWACF-140	90.68	8.29	0.72	0.21	0.10	0.091
LWACF-220	92.29	7.02	0.45	0.14	0.09	0.076

## Data Availability

The data presented in this study are available on request from the corresponding author.

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
