# Peer review of "Synthesis of Microporosity Dominant Wood-Based Activated Carbon Fiber for Removal of Copper Ions"

_polymers, 2022, doi:10.3390/polym14061088_

Round 1
Reviewer 1 Report
Мой отзыв вы можете найти в прикрепленном файле.
Reviewer Report
Manuscript Number: 1584815
Title: Synthesis of microporosity dominant wood-based activated carbon fiber for removal of copper ions
Zhi Jin , Zhen Zeng, Shenghui Hu, Lina Tang, Yuejin Fu, Guangjie Zhao
The manuscript deals with obtaining of microporosity activated carbon fiber (LWACF) for adsorption removal of copper ions. The surface properties, pore structure and morphology of LWACF were investigated using XPS measurements, scanning electron microscopy, and Brunauer-Emmett-Teller surface area measurements. The effect of adsorbent dosages on the adsorption efficiency of copper ions removal was investigated.
The study is interesting from the ecological point of view. Overall, my evaluation is positive, and I believe that the manuscript can be published in Polymers if the questions and comments below will be considered and accounted before publication of this paper to reach the standard quality of the journal.
Questions and comments:
Line 48-49: It should be specified what size (radius or diameter?) and what particle (atom, unhydrated or hydrated ion?).
Line 99-100: The initial concentration of the copper solution is not indicated, but only the concentration after dilution is indicated. It is necessary to interpret why such a pH value was chosen. Is there wastewater (which contain copper ions) with pH =5.5?
Line 101-104: The methodology for kinetic studies is indicated, but kinetic studies are not included in the manuscript.
Line 107-109: It should be indicated the conditions for determination of copper ions
Line 179-183: “It is commonly believed that there were five primary mechanisms of heavy metals by porous carbon materials from aqueous solutions, i.e., complexation, cation exchange, precipitation, electrostatic interactions, and chemical reduction [28-30].” The authors listed the known mechanisms but did not consider the predominant mechanisms for the new adsorbent.
In conclusion, authors paid attention to the preparation of the adsorbent and some of its characteristics. Important information for adsorption studies is the pH point of zero charge of the adsorbent. In addition, there are no adsorption studies about effect of pH, concentration of copper ions and kinetic studies. This information allows in full to evaluate adsorption properties of the new adsorbent.
Author Response
Thank you very much for your letter and the reviewers’ comments concerning our article. The comments are all valuable and very helpful for improving our research. We have studied the comments carefully and made corrections which we hope meet with approval. Revised portion are marked in red in the revised manuscript. The main corrections and the responses to the reviewer’ comments are as following:
Line 48-49: It should be specified what size (radius or diameter?) and what particle (atom, unhydrated or hydrated ion?).
Replying: The diameter of unhydrated Cu(II) is 0.144 nm.
Line 99-100: The initial concentration of the copper solution is not indicated, but only the concentration after dilution is indicated. It is necessary to interpret why such a pH value was chosen. Is there wastewater (which contain copper ions) with pH =5.5?
Replying: We have modified the description in the revised manuscript. In the present work, the Cu(II) began to deposit at pH =6.0, so we prepared Cu(II) solution with pH of 5.5.
Line 101-104: The methodology for kinetic studies is indicated, but kinetic studies are not included in the manuscript.
Replying: Thank you very much for your all valuable and very helpful comments. In the present work, we mainly focused on the synthesis and characterization of the LWACF, We have modified the description in the revised manuscript. In the following work, we will systematically investigate the recyclability and the kinetic analysis of the prepared adsorbent.
Line 107-109: It should be indicated the conditions for determination of copper ions
Replying: We have added the conditions for determination of copper ions in the revised manuscript.
Line 179-183: “It is commonly believed that there were five primary mechanisms of heavy metals by porous carbon materials from aqueous solutions, i.e., complexation, cation exchange, precipitation, electrostatic interactions, and chemical reduction [28-30].” The authors listed the known mechanisms but did not consider the predominant mechanisms for the new adsorbent.
Replying: We have modified the mechanism discussion in the revised manuscript.
In conclusion, authors paid attention to the preparation of the adsorbent and some of its characteristics. Important information for adsorption studies is the pH point of zero charge of the adsorbent. In addition, there are no adsorption studies about effect of pH, concentration of copper ions and kinetic studies. This information allows in full to evaluate adsorption properties of the new adsorbent.
Replying: We have modified the description in the revised manuscript.

Reviewer 2 Report
After reading the manuscript entitled:"Synthesis of microporosity dominant wood-based activated carbon fiber for removal of copper ions" which is reporting the synthesis under different treatment conditions and final applications of samples for removal of copper. The manuscript is interesting for a wide range of researchers. However, it is a short manuscript, more experimental results related to the application must be provided, accordingly, I recommend the following to be done:
1- Since economical issues are an important factor for environmental applications, it is very important to have good recyclibility of an adsorbent to be used commercially.
2- Kinetic analysis is also important since the time required to achieve equilibrium indicated how fast the treatment could be.
3- The main challenge in removing heavy metals, usually when we deal with very low concentrations. A good adsorbent is an adsorbent with high sensitivity even at low concentrations (less than 20 ppm). I suggest running some experiments at low concentrations.
Author Response
Thank you very much for your letter and the reviewers’ comments concerning our article. The comments are all valuable and very helpful for improving our research. We have studied the comments carefully and made corrections which we hope meet with approval. Revised portion are marked in red in the revised manuscript. The main corrections and the responses to the reviewer’ comments are as following:
1- Since economical issues are an important factor for environmental applications, it is very important to have good recyclibility of an adsorbent to be used commercially.
Replying: Thank you very much for your all valuable and very helpful comments. In the present work, we mainly focused on the synthesis and characterization of the LWACF, We have modified the description in the revised manuscript. In the following work, we will systematically investigate the recyclability and the kinetic analysis of the prepared adsorbent.
2- Kinetic analysis is also important since the time required to achieve equilibrium indicated how fast the treatment could be.
Replying: In the present work, we mainly focused on the synthesis and characterization of the LWACF and we have modified the description in the revised manuscript. In the following work, we will systematically investigate the kinetic analysis of the prepared adsorbent.
3- The main challenge in removing heavy metals, usually when we deal with very low concentrations. A good adsorbent is an adsorbent with high sensitivity even at low concentrations (less than 20 ppm). I suggest running some experiments at low concentrations.
Replying: In the present work, we mainly focused on the synthesis and characterization of the LWACF and we have modified the description in the revised manuscript. In the following paper We will systematically investigate the adsorption performance of the adsorbents.

Reviewer 3 Report
Sample preparation and activation part of the manuscript is sound in the sense that provides necessary details about the synthesis conditions and relevant BET, PSD and XPS results. However, discussion of adsorption, especially "kinetic" part, as authors put it, is lacking. There is no graph representing the uptake over time (especially at low reaction times) that would be needed for kinetic evaluation. How fast saturation is achieved after which comparison of dosages would be relevant? No fitting to connect results from Cu2+ uptake to BET, PSD and XPS, just a vague speculation. Mathematically stated correlation would be helpful in this regard. Almost no comparison of uptake to literature. Possible mechanisms to reuse the material?
"high-ly-efficient " highly , in abstract.
Author Response
Thank you very much for your letter and the reviewers’ comments concerning our article. The comments are all valuable and very helpful for improving our research. We have studied the comments carefully and made corrections which we hope meet with approval. Revised portion are marked in red in the revised manuscript. The main corrections and the responses to the reviewer’ comments are as following:
Sample preparation and activation part of the manuscript is sound in the sense that provides necessary details about the synthesis conditions and relevant BET, PSD and XPS results. However, discussion of adsorption, especially "kinetic" part, as authors put it, is lacking. There is no graph representing the uptake over time (especially at low reaction times) that would be needed for kinetic evaluation. How fast saturation is achieved after which comparison of dosages would be relevant? No fitting to connect results from Cu2+ uptake to BET, PSD and XPS, just a vague speculation. Mathematically stated correlation would be helpful in this regard. Almost no comparison of uptake to literature. Possible mechanisms to reuse the material?
Replying: Thank you very much for your all valuable and very helpful comments. In the present work, we mainly focused on the synthesis and characterization of the LWACF. In the following paper We will systematically investigate the adsorption performance of the adsorbents. We have polished the title and abstract in the revised manuscript.
"high-ly-efficient " highly , in abstract.
Replying: We have modified the description in the revised manuscript.
Round 2
Reviewer 1 Report
You can find reviewer's comments towards your reply in the attached file
The Authors considered some reviewer's comments, which were made in the original version, however they didn't take into account the reviewer's comments about absent important experimental adsorption data and their interpretation in the revised manuscript. Unfortunately, for now, I evaluate the revised version as unsuitable for publication. In my opinion, the paper can be suitable for publication only after accounting for the following comments:
- It should be corrected Cu(II) ion to Cu2+ ion in the manuscript.
- Why was used diameter of unhydrated Cu2+ ion? It is well known that in aqueous solution there are hydrated ions. It should take into consideration when we study adsorption from aqueous solution that sizes of hydrated and unhydrated ions are different.
- Is there wastewater (which contains copper ions) with pH =5.5? The authors did not answer this question.
- Line 96: “For kinetic studies,…” The methodology for kinetic studies is indicated, but kinetic curves did not include in the manuscript.
- In Section 3.4 adsorption studies of Cu2+ are insufficient presented and they should be supplemented.
- Important information for adsorption studies is the pH point of zero charge of the adsorbent. There are no adsorption studies about effect of pH, concentration of copper ions and kinetic curves in manuscript. This information allows in full to evaluate adsorption properties of the new adsorbent. The authors did not consider this comment.
Author Response
Reviewer’s comments
Line 48-49: It should be specified what size (radius or diameter?) and what particle (atom, unhydrated or hydrated ion?).
Replying: In the present work, CuSO4·5H2O was used to prepare the standard solutions, so the diameter of hydrated Cu(II) is 0.144 nm.
Line 99-100: The initial concentration of the copper solution is not indicated, but only the concentration after dilution is indicated. It is necessary to interpret why such a pH value was chosen. Is there wastewater (which contain copper ions) with pH =5.5?
Replying: We have modified the description in the revised manuscript. In the present work, the Cu2+ began to deposit at pH =6.0, so we prepared Cu(II) solution with pH of 5.5. We do not use the wastewater.
Line 101-104: The methodology for kinetic studies is indicated, but kinetic studies are not included in the manuscript.
Replying: Thank you very much for your all valuable and very helpful comments. In the present work, we mainly focused on the synthesis and characterization of the LWACF, We have modified the description in the revised manuscript. In the following work, we will systematically investigate the recyclability and the kinetic analysis of the prepared adsorbent.
Line 107-109: It should be indicated the conditions for determination of copper ions
Replying: We have added the conditions for determination of copper ions in the revised manuscript.
Line 179-183: “It is commonly believed that there were five primary mechanisms of heavy metals by porous carbon materials from aqueous solutions, i.e., complexation, cation exchange, precipitation, electrostatic interactions, and chemical reduction [28-30].” The authors listed the known mechanisms but did not consider the predominant mechanisms for the new adsorbent.
Replying: We have modified the mechanism discussion in the revised manuscript.
In conclusion, authors paid attention to the preparation of the adsorbent and some of its characteristics. Important information for adsorption studies is the pH point of zero charge of the adsorbent. In addition, there are no adsorption studies about effect of pH, concentration of copper ions and kinetic studies. This information allows in full to evaluate adsorption properties of the new adsorbent.
Replying: We have modified the description in the revised manuscript.

Reviewer 2 Report
Accept.
Author Response
We have polished the revised manuscript.
Reviewer 3 Report
Authors have not answered to any of the raised issue. They state that adsorption studies will be done in another manuscript. Therefore, I advise against publication in present form. If authors limit the manuscript to the preparation only, without brushing over the adsorption in one passage manuscript might be reconsidered.
Author Response
We have modified the discussion in the revised manuscript.
Round 3
Reviewer 3 Report
Authors have redesigned the text somewhat to accent the main point of the work and can now be published.